# Enhancing Milk Production by Nutrient Supplements: Strategies and Regulatory Pathways

**DOI:** 10.3390/ani13030419

**Published:** 2023-01-26

**Authors:** Fengguang Pan, Peizhi Li, Guijie Hao, Yinuo Liu, Tian Wang, Boqun Liu

**Affiliations:** 1Laboratory of Nutrition and Functional Food, College of Food Science and Engineering, Jilin University, Changchun 130062, China; 2Key Laboratory of Healthy Freshwater Aquaculture, Ministry of Agriculture and Rural Affairs, Huzhou 313001, China; 3Key Laboratory of Fish Health and Nutrition of Zhejiang Province, Zhejiang Institute of Freshwater Fisheries, Huzhou 313001, China; 4Key Laboratory of Genetics and Breeding, Zhejiang Institute of Freshwater Fisheries, Huzhou 313001, China; 5Department of Laboratory Animals, College of Animal Sciences, Jilin University, Changchun 130062, China

**Keywords:** nutrients, milk protein, milk fat, regulatory pathway, mTORC

## Abstract

**Simple Summary:**

The demand for the milk and dairy products keeps growing due to the growing population, the rising consumption of protein-enriched food, and other factors. Thus, enhancing milk production becomes a critical issue for the world. Of all the factors related to promoting milk production, nutrient supplementation is widely investigated. First, researchers provided mammary epithelial cells or animals with appropriate nutrients such as amino acids, peptides, lipids, carbohydrates, and others. Then, they detected the increase in milk protein expression or milk production. Finally, the researcheres further investigated the essential pathways on which the nutrients act. As much effort has been made on this topic, we summarize recent work on enhancing milk production through nutrient supplements. We hope this review provides a systematic reference for researchers in related fields and can help understand the mechanism of milk production promotion by nutrients.

**Abstract:**

The enhancement of milk production is essential for dairy animals, and nutrient supplements can enhance milk production. This work summarizes the influence of nutrient supplements—including amino acids, peptides, lipids, carbohydrates, and other chemicals (such as phenolic compounds, prolactin, estrogen and growth factors)—on milk production. We also attempt to provide possible illuminating insights into the subsequent effects of nutrient supplements on milk synthesis. This work may help understand the strategy and the regulatory pathway of milk production promotion. Specifically, we summarize the roles and related pathways of nutrients in promoting milk protein and fat synthesis. We hope this review will help people understand the relationship between nutritional supplementation and milk production.

## 1. Introduction

Dairy production is an essential economic characteristic of dairy animals. Milk and dairy products are consumed all over the world. In 2021, the world’s milk production reached nearly 928 million tons [1], and the demand will continue to increase [2]. It is predicted that world milk production will grow at an annual rate of 1.7%. Milk production will reach 1.020 billion tons by 2030, faster than most other major agricultural products [3]. However, the average milk production per cow per lactation period varies from 1.3 to 10 tons. For example, the average milk production of dairy cows in Saudi Arabia and Israel is more than 10 tons, while in India, it is only 1.3 tons [4]. Though genetics greatly influence milk production, nutrient supplements still play an essential role.

In recent years, we have taken traditional and molecular breeding efforts, especially in breed registration, productivity determination, body size identification, artificial insemination, genome-wide association analysis (GWAS) and somatic cell cloning, achieving significant increases in milk production in dairy cows [5,6,7,8,9,10]. For example, in developing countries such as India, a Holstein cow’s average daily milk production increased from 1259 kg in 2009–2010 to 1700 kg in 2018–2019 [11]. However, it is becoming increasingly difficult to further improve milk quality and yield traits through conventional and molecular breeding methods. 

A promising and upcoming strategy for improving milk production and quality in developing countries is providing high-quality feed and balanced nutrition [12]. The strategy involves adding appropriate nutrients (amino acids, peptides, lipids) to the cell culture medium or feed that promotes milk quality and production. This work summarizes recent studies on nutrients enhancing milk production and relevant pathways to mammal milk production.

## 2. The Enhancement of Milk Production by Amino Acids and Peptides

### 2.1. Amino Acid

The most fundamental pathway for amino acids to enhance milk synthesis is the mTOR pathway (Table 1). The mammalian target of rapamycin (mTOR) is an essential signaling factor widely distributed in mammalian cells. It is essential in cell growth and protein synthesis [13]. Phosphatidylinositol-3 Kinase (PI3K) exists in the cytoplasm and has dual protein kinase and phospholipid kinase activities. PI3K is eventually converted to phosphatidylinositol 3,4,5-triphosphate (PIP3) after activation [14,15]. The serine/threonine kinase AKT, also known as protein kinase B (PKB), is a key downstream protein of PI3K. It exists in the cytoplasm, and they play a vital role in regulating cell growth, proliferation, survival, and metabolism [16,17]. The PI3K-Akt pathway is upstream of the mTOR signaling pathway, so factors that stimulate this pathway can also activate the mTOR pathway (Figure 1). PIP3 can bind to the PH domain of AKT proteins as a second messenger when PI3K is activated. The activated AKT is transferred from the cell membrane to the cytoplasm or nucleus and then continues to target and regulate downstream signaling molecules-mTOR [18]. Therefore, it further regulates downstream proteins after mTOR signaling is activated through the PI3K/AKT pathway. The phosphorylated mTOR can activate ribosomal protein S6 kinase 1 (S6K1) and eukaryotic translation initiation factor 4e-binding protein 1 (4EBP1), promoting their involvement in translation and protein synthesis [19].

It has been reported that amino acid addition activates the mTORC1 pathway via the Septin6 factor in dairy bovine mammary epithelial cells (BMECs), promoting cell growth and milk protein synthesis [20]. Amino acid supplementation in BMECs can also stimulate mTOR signaling through Seryl-tRNA synthetase (SARS) and Glycyl-tRNA synthetase (GlyRS), thereby exerting positive effects on cell proliferation and casein synthesis [21,22]. In addition, the optimal ratio of essential amino acids added to BMECs, and bovine mammary tissue explants (MTE) can stimulate β-casein synthesis [23]. Among all the amino acids, researchers focus on several amino acids, including methionine, leucine, valine and lysine (Figure 1). We review these works in the following part.

#### 2.1.1. Methionine

Among all the amino acids, methionine attracts the most attention (Table 1). The methionine supply can increase milk protein synthesis by increasing the expression of related proteins in early lactation [24,25]. In addition, methionine can promote milk fat synthesis by promoting the mTORC1 signaling pathway. Amino-acid transporter 2 (ASCT2) is involved in SARS-mediated β-casein synthesis in BMECs via the mTOR signalling pathway [26]. Methionine positively regulates milk protein and fat synthesis in BMECs through the sodium-coupled neutral amino acid transporter 2 (SNAT2)-PI3K signaling pathway [27]. The PI3K-Akt-mTOR signaling pathway can regulate lipid synthesis by regulating the expression of the lipid synthesis-related factor sterol response element-binding protein (SREBP1) and its related esterases [28]. SREBP-1c acts as a transcription factor that controls the expression of lipid synthesis genes. Therefore, some factors activating SREBP-1c can also stimulate milk fat synthesis. For example, fatty acid binding protein 5 (FABP5), an intracellular lipid transporter, has been confirmed to be a crucial regulator of methionine- and estrogen-induced SREBP-1c gene expression in BMECs [29]. Besides, nuclear receptor co-activator 5 (NCOA5) and glucose-regulated protein 78 (GRP78) can also act as regulators of methionine-stimulated mTOR expression and activation [30,31]. They both function upstream of mTOR.

Supplementing 2.5 g/d of rumen-protected methionine per lactating Saanen goat led to significantly increased milk production [32]. The addition of methionine at 42 g/d per Holstein cow positively affected milk yield [33]. Studies have shown that adding microencapsulated DL-Met (0, 11.0, 19.3 and 27.5 g/d) to diets deficient in metabolizable methionine (10 g/d) can also increase milk production. It was confirmed that the increase rate of milk protein yield was the highest with the addition of 19.3 g/d methionine [34]. Holstein cows fed rumen-protected methionine increased milk production (from 1.43 kg/d to 1.48 kg/d) [35]. Studies have shown that milk production increases when Holstein cows are fed on methionine 10 g/day in their corn grain and soybean meal concentrate diets [36].

#### 2.1.2. Leucine

Leucine affects the expression of mTORC1 through system L and system A transporters such as L-type amino acid transporter 1 (LAT1) and SNAT2 [37]. It has been confirmed that mTOR is a target molecule of leucine. Leucine promotes mTORC1 expression in cells through the Sestrin2/GATOR2/GATOR1 pathway [38]. Meanwhile, leucine can promote cellular processes through the PI3K/AKT/mTOR pathway as a signaling molecule [39]. It has been shown that leucine activates PI3K, thereby upregulating DEAD-Box Helicase 59 (DDX59). DDX59 positively regulates the mTOR and SREBP-1c signaling pathways leading to milk synthesis in primary BMECs [40]. Guanine nucleotide-binding protein subunit gamma-12 (GNG12) also mediates leucine regulation of mTORC1 signaling. It activates the mTORC1 pathway by interacting with the regulator, which can promote cow mammary epithelial cells (CMECs) growth and casein synthesis [41]. The addition of leucine to primary BMECs from mid-lactation Holstein cows promotes milk protein synthesis by affecting mTOR upstream factors NCOA5 and GRP78 [30,31]. 

Leucine can promote milk protein synthesis by the JAK2 (Janus kinase 2) and STAT5 (signal transducers and activators of transcription 5) pathways [42]. During gland development, JAK-STAT is essential for the survival of mammary gland secretory epithelial cells. The JAK-STAT pathway consists of three parts: tyrosine kinase-related receptors that receive signals, JAK that transmit signals, and STAT that produce effects [43]. When cytokines specifically bind to the corresponding receptors, the receptor molecules can be induced to dimerize. JAK kinases can interact closely with receptors and activate themselves. Following activation of the kinase, the receptor’s tyrosine residues can be stimulated and phosphorylated. After kinase phosphorylation, it can be recognized and bound by STAT. As a result, STAT tyrosine residues are phosphorylated to form dimers, enter the nucleus, bind to the target gene’s DNA regulatory region, regulate the target gene’s transcription, and produce corresponding biological effects [44]. Studies have confirmed that acetate, leucine and their interactions affect milk protein synthesis through JACK2/STAT5, mTOR and AMPK pathways in BMECs [42].

#### 2.1.3. Valine

Valine can regulate milk protein synthesis [45]. Studies have confirmed that valine promotes milk protein synthesis in porcine mammary epithelial cells, and the regulatory pathways are mTOR, Ras/ERK and AKT [46,47]. The extracellular signal-regulated kinase (ERK) is a member of mitogen-activated protein kinase (MAPK) family. In the ERK-MAPK pathway, Ras GTPase is an upstream activator in the cascade response of MAPK (Ras-Raf-MEK-ERK). The Ras and ERK-MAPK pathways can be activated by growth factors, polypeptide hormones, neurotransmitters, etc [48]. Studies have shown that treatment with L-valine (L-enantiomer of valine) to porcine mammary epithelial cells can promote the expression of mTOR, Ras and ERK1/2 [46], thus playing a pivotal role in milk protein synthesis. The milk yield of each prolific late lactating Holstein cow fed of valine 40 g/d could reach 25.2 kg/d, which was higher than that of the control group (22.0 kg/d) [49]. It has also been demonstrated in lactating sows that adding 3.07% branched-chain amino acids to the diet can increase milk production by 21% [50].

#### 2.1.4. Lysine

Lysine is the first limiting amino acid for milk protein synthesis in dairy cows. In the process of lysine-stimulated milk protein, GPRC6A-PI3K-FABP5 regulatory axis plays an important role [51]. GPRC6A (G protein-coupled receptor family C group 6 member A), a widely expressed G-protein coupled receptor, is a significant regulator of energy metabolism, cell proliferation and differentiation. Lysine also regulates milk protein synthesis through the mTOR and JAK2-STAT5 pathways. It has been shown that 1.0 mmol/L lysine stimulation significantly increases the expression of amino acid transporter B^0,+^ (ATB^0,+^), STAT5, and mTOR in BMECs [52]. Lysine promotes β-casein synthesis in BMECs via the SLC6A14-ERK-CDK1-mTOR signaling pathway [53]. Lysine also promotes mammary gland development in mice at puberty through the PI3K/AKT/mTOR signaling pathway [54]. The optimal ratio of supplemented Lys/Met is 3:1. This ratio can also regulate casein synthesis in BMECs, mainly through JAK2/ELF5, mTOR and its downstream ribosomal protein S6 kinase B1 (RPS6KB1) and eukaryotic translation initiation factor 4E binding protein 1 (EIF4EBP1) signaling pathways [55].

#### 2.1.5. Other Amino Acids

Other amino acids can also have a positive effect on milk production (Table 1). Arginine supply affects mTORC1 expression and AMPK pathway, thereby affecting α-s1-casein synthesis in BMECs [56]. Histidine and threonine can promote milk protein synthesis through the mTOR pathway [57,58,59]. Taurine, 2-aminoethanesulfonic acid, can regulate insulin secretion, and the pathways that have been confirmed are PI3K/AKT, AKT/FOXO1, JAK2/STAT3 and mTOR/AMPK [60,61]. Taurine also plays an essential role in milk protein synthesis. Studies have shown that taurine promotes milk synthesis through the GPR87-PI3K-SETD1A signaling pathway [62]. GPCRs can sense amino acid signals to activate downstream pathways [63,64]. Therefore, taurine stimulates the expression of GPR87 to activate PI3K, activating the downstream signaling SET domain containing 1A (SETD1A) of the pathway.

Each lactating Holstein cow was fed 10 mg/kg 5-aminolevulinic acid for 14 days, and milk protein content increased [65]. Studies have shown that the continuous addition N-carbamoylglutamate (NCG) of 20 g/d per cow to the feed from 4 weeks before calving to 10 weeks after calving improves the lactation performance of postpartum cows [56]. Previous studies have demonstrated that feeding lactating sows (Large White) with 1% glutamine for 14 days has a positive effect on milk production (+ 3.34 kg) [66]. Supplementation with 1.2 g/day of rumen-protective gamma-aminobutyric acid increased feed intake and milk protein production in Holstein cows [67]. The addition of three rumen-protected methionines (Met, Lys, and His) have the potential to increase milk production [68].

We can conclude that adding amino acids to the feed has a specific positive effect on milk yield. In recent years, researchers focused on exploring the functional genes and key pathways related to the promotion of milk production at the cellular level. Amino acids can positively affect the proliferation and milk protein synthesis of mammary epithelial cells. After amino acids are transported into cells, they stimulate the expression of certain genes involved in milk protein synthesis. The mTOR is involved in multiple biological processes (Figure 1). Septin6, SARS, GlyRS, DDX59, and GNG12 are the upstream genes of mTOR pathway. After being stimulated by amino acids, this pathway will be activated, thereby promoting milk protein synthesis. The PI3K/AKT is the upstream pathway of mTOR, so certain amino acids exert their effects by stimulating upstream factors of PI3K, such as NCOA5. In addition, leucine and lysine can also affect the JAK2/STAT5 pathway and thus affect milk protein synthesis.

**Table 1 animals-13-00419-t001:** Effects of amino acids on milk production and its potential regulatory pathways.

Items	Treatments	Functions	Potential Signaling Pathways	References
Amino acid (+)	Dairy cow mammary epithelial cells	Cell growth ↑ Casein synthesis ↑	Septin6-mTOR signaling pathway	[20]
Essential amino acids (+)	Bovine mammary epithelial cells	Cell proliferation ↑ β-casein production ↑	SARS- mTOR signaling pathway	[22]
Amino acid (+)	Bovine mammary epithelial cells	Milk protein ↑ Fat synthesis ↑	GlyRS-mTOR signaling pathway	[21]
Essential amino acids (+)	Bovine mammary epithelial cell line, mammary tissue explants	β-casein expression ↑	mTOR signaling	[23]
Methionine (+)	Cow mammary gland tissue	Milk protein synthesis ↑	AKT phosphorylation	[24]
Methionine (+)	Bovine mammary epithelial cells (0.6 mmol/L)	Milk protein ↑ Fat synthesis ↑ Cell proliferation ↑	SNAT2-PI3K signaling pathway	[27]
Methionine (+)	Bovine mammary epithelial cells (0.6 mmol/L)	Cell growth ↑ β-casein synthesis ↑ ASCT2 expression ↑	ASCT2/SARS/mTOR signaling pathway	[28]
Methionine (+)	Bovine mammary epithelial cells	Lipid droplet formation ↑ β-casein ↑	FABP5-SREBP-1c signaling pathway	[29]
Methionine (+)	Lactating Saanen goats	Milk production ↑ Milk protein ↑	——	[32]
Methionine (+)	Holstein cows	Milk yield ↑	——	[33]
Methionine (+)	Holstein cows	Milk production ↑ Milk protein ↑	——	[34]
Methionine (+)	Holstein cows	Milk protein production ↑	——	[35]
Methionine (+)	Holstein cows	Milk production ↑	——	[36]
Leucine (+)	Bovine mammary epithelial cells (0.75 mmol/L)	Milk protein ↑ Milk fat synthesis ↑	PI3K-DDX59 signaling pathway	[40]
Leucine (+)	Cow mammary epithelial cells	Cell growth ↑ Casein synthesis ↑	GNG12-mTORC1 signaling pathway	[41]
Leucine and methionine (+)	Bovine mammary epithelial cells	Milk Fat ↑	mTOR-CRTC2-SREBP-1c signaling pathway	[69]
Leucine and methionine (+)	Bovine mammary epithelial cells	mTOR phosphorylation ↑ β-casein synthesis ↑	NCOA5- PI3K-mTOR signaling pathway	[31]
Leucine and methionine (+)	Bovine mammary epithelial cells	Milk synthesis ↑ Cell proliferation ↑	AnxA2 PI3K-mTOR-SREBP-1c/Cyclin D1 signaling pathway	[70]
Leucine and methionine (+)	Bovine mammary epithelial cells	β-casein, triglycerides, Lactose synthesis ↑ Cell viability ↑ Cell proliferation ↑	mTOR-SREBP-1c signaling pathway	[71]
Leucine and methionine (+)	Bovine mammary epithelial cells	Milk protein ↑ Milk fat ↑ Cell proliferation ↑	GRP78-mTOR signaling pathway	[30]
Leucine, acetate, and their interaction (+)	Bovine mammary epithelial cells (1.8 mmol/L Leucine or 8–10 mmol/L acetate)	Milk protein ↑	JACK2/STAT5, mTOR, and AMPK pathway	[42]
Valine	Primiparous gilts (total lysine: lysine = 0.93:1)	Milk fat synthesis ↑	——	[45]
L-Valine (+)	Porcine mammary epithelial cells (0.5 mmol/L)	Cell numbers ↑ Protein synthesis ↑	mTOR and Ras/ERK signaling pathways	[46]
Valine (+)	Porcine mammary epithelial cells	Fatty acids synthesis ↑ Intracellular triacylglycerol content ↑	AKT-mTOR-SREBP1 signaling pathway	[47]
Valine (+)	Holstein cows	Milk production ↑	——	[49]
Branched-chain amino acids (+)	Multiparous sows (Yorkshire × Landrace)	Milk production ↑	——	[50]
Lysine (+)	Bovine mammary epithelial cells (0.70 mmol/L)	Cells numbers ↑ Milk fat synthesis ↑	GPRC6A-PI3K-FABP5 signaling pathway	[51]
Lysine (+)	Bovine mammary epithelial cells (1.0 mmol/L)	Protein synthesis ↑	ATB^0,+^,mTOR and JAK2-STAT5 pathways	[52]
Lysine (+)	Bovine mammary epithelial cells (0.70 mmol/L)	β-casein synthesis ↑	SLC6A14-ERK-CDK1-mTOR signaling pathway	[53]
Lysine (+)	Mouse mammary epithelial cells	Cell proliferation ↑	PI3K/AKT/mTOR signal axis	[54]
Lysine, methionine (+)	Bovine mammary epithelial cells (Lys/Met ratio = 3:1, 1.2 mmol/L Lys, 0.4 mmol/L Met)	Casein biosynthesis ↑	JAK2/ELF5, mTOR, and its downstream RPS6KB1 and EIF4EBP1 signaling	[55]
Methionine and arginine (+)	Bovine mammary epithelial cells	α-s1-casein abundance ↑	mTOR signaling; AMPK pathways	[25]
Leucine or arginine (-)	Mid-lactation Holstein cows (5 day continuous Leucine-163 g/d; arginine-158 g/d)	Milk yield ↓ Milk protein yield ↓	——	[72]
N-carbamoylglutamate (+)	Holstein cows (20 g/d per cow, n = 15)	Milk production ↑	——	[56]
5-aminolevulinic acid (+)	Holstein cows (10 mg/kg per cow)	Milk protein ↑; Milk casein contents ↑	——	[65]
Glutamine (+)	Lactating sows (Large White) (1%)	Milk yield ↑	——	[66]
Taurine (+)	Bovine mammary epithelial cells	Milk protein ↑ Fat synthesis ↑ mTOR phosphorylation ↑ SREBP-1c expression ↑	GPR87-PI3K-SETD1A signaling pathway	[62]
Leucine and histidine (+)	Immortalized bovine mammary epithelial cell (0.45–10.80 mmol/L Leucine 0.15–9.60 mmol/L histidine	Milk protein ↑	mTOR signaling pathway	[59]
Histidine, lysine, methionine, leucine	Immortalized bovine mammary epithelial cell line (His: Lys: Met: Leu = 5:6:1:7)	β-casein expression ↑	mTOR signaling pathway	[57]
Threonine, isoleucine, valine, leucine	Immortalized bovine mammary epithelial cell line (Lysine: valine = 1.12:1 when Lysine: methionine is ideal)	β-casein expression ↑	mTOR signaling pathway	[58]
Rumen-protected-methionine, lysine, histidine (+)	Holstein cows	Milk production ↑	——	[68]
Rumen-protected gamma-aminobutyric acid	Holstein dairy cows	Milk protein yield ↑	——	[67]

↑: increase; ↓: decrease; ——: reported but not fully confirmed.

### 2.2. Peptides

Peptides are formed by the dehydration condensation of two or more amino acids. It is involved in hormones, nerves, cell growth and reproduction and is highly significant for biologically active functions [73]. The proton-coupled oligopeptide transporter family (SLC15) plays an essential role in intracellular di- and tri-peptide uptake. The SLC15 family consists of four members, including PepT1 (SLC15A1), PepT2 (SLC15A2), Pht1 (SLC15A4) and Pht2 (SLC15A3) [74], of which both PepT1 and PepT2 play crucial roles in the mammary gland [75]. 

Peptides can enhance mammalian (bovine, goat) milk protein and affect its potential signaling pathway. Among them, mTOR still plays an important role. It has been confirmed that ghrelin, kisspeptin-10, dipeptide (methionyl-methionine), and substrate (threonyl-phenylalanyl-phenylalanine) are of great significance for promoting mammalian milk fat synthesis (Table 2). Ghrelin is a polypeptide hormone containing 28 amino acids isolated from the stomach. It is available in both acylated ghrelin (AG) and unacylated ghrelin (UAG) forms [76]. Ghrelin stimulates β-casein expression in cultured goat primary mammary epithelial cells [77]. In BMECs, both AG and UAG act on ERK1/2 and AKT signaling pathways to promote CSN2 expression [78]. ERKs are closely related to cell growth, proliferation, and differentiation [79]. Kisspeptin-10 is a neuropeptide hormone, a short peptide formed by the cleavage of Kisspeptin-54 [80]. A study has shown that 100 nmol/L of Kisspeptin-10 promotes β-casein synthesis in Holstein cow mammary epithelial cells through GPR54 and its downstream signaling pathways mTOR, ERK1/2, STAT5 and AKT [81]. Kisspeptins are the products of the Kiss-1 gene, and the Kiss-1/GPR54 system is expressed in mammalian mammary glands [82,83], thereby promoting downstream signaling. Dipeptide (Methionyl-Methionine) positively affects JAK2-STAT5 and mTOR signaling pathways in BMECs after being absorbed by PepT2, thereby promoting β-casein synthesis and cell proliferation [84]. Studies also showed that after stimulation of BMECs with tripeptides and lactogenic hormones, the abundance of PepT1 and PepT2 can be increased, thereby promoting milk fat synthesis [85,86]. In addition, it has been shown that OPH3-1, a fraction of the octopus protein hydrolysis product bioactive peptide, promotes mammary epithelial cell proliferation and casein synthesis in mice [87].

Compared with amino acids, there are relatively few studies on peptides promoting milk production. We find that the mechanism of peptide promoting milk production is similar to amino acids. The mTOR and JAK2-STAT5 signaling pathways are also important in peptide-promoted milk fat synthesis. In these pathways, related factors such as GPR54, ERK1/2 and AKT all affect milk fat synthesis. They will increase the expression of β-casein and PepT in mammary epithelial cells through the pathway and finally achieve the effect of promoting lactation.

## 3. The Enhancement of Milk Production by Lipids

Lipids include triglycerides and lipoids (phospholipids, sterols). Fatty acids are an essential source of energy and affect cellular and tissue metabolism, among others [88]. The mTOR pathway plays an important role in the promotion of milk fat synthesis by fatty acids. In addition, some studies have confirmed the positive effect of JAK2-STAT5 (Table 3). It is affected by many fatty acid-related factors in milk fat synthesis, such as SREBP1, peroxisome proliferators-activated receptor gamma (PPARG), fatty acid binding protein 3 (FABP3), etc. SREBPS is a significant factor regulating fat synthesis in animals. It regulates the activities of lipogenesis-related enzymes by regulating the gene transcription of lipogenesis-related enzymes, thereby controlling fat synthesis [89]. It has been shown that overexpression of SREBP1 promotes milk fat synthesis in cow and goat mammary epithelial cells [90,91]. PPAR can protect cells from oxidative damage, regulate glucose and lipid metabolism, and maintain the balance of inflammation in the body [92,93,94]. PPARG has an important link with the transcriptional regulation of fatty acids in the mammary gland [95]. FABP is a member of the intracellular lipid-binding protein superfamily. It plays a significant role in intracellular long-chain fatty acid uptake, transport and metabolism [96]. FABP3, one of the target genes of SREBP1, is the main protein for the rapid diffusion of long-chain fatty acids and the selective targeting of specific organelles for metabolism.

We can still find that researchers are exploring the effects of fatty acids on pathways related to milk fat synthesis at the cellular level (Table 3). Short-chain fatty acids such as acetic acid can promote BMECs proliferation and milk fat synthesis. Adding 6 mmol/L acetic acid to BMECs can stimulate the expression of fatty acid-related genes (such as SREBP1, mTOR, FABP3, etc.), and it is proved that the pathway is mTOR/eIF4E [97]. Acetic acid and β-hydroxybutyric acid work together to promote dairy cow mammary epithelial cells (DCMECs) triglyceride synthesis [98]. Studies of long-chain saturated and unsaturated fatty acids on milk fat synthesis have shown that saturated fatty acids, especially C18:0, can regulate milk fat synthesis in goats via PPARG [99]. Some C18 unsaturated fatty acids (e.g., oleic, linoleic, linolenic, palmitic, stearic, and palmitic) have also been identified. Oleic acid, stearic acid and palmitic acid can increase the accumulation of lipid droplets in BMECs by affecting the expression of FABP3, thereby upregulating SREBP1 and PPARG to promote milk fat synthesis [100]. Different ratios of unsaturated fatty acids (oleic, linoleic, and linolenic) can affect the synthesis of related proteins in BMECs [101]. Adding volatile fatty acids to Holstein dairy cattle feed can also increase the mRNA expression of PPARG, SREBPF1, FABP3 and other genes, and the optimal addition amount is 60 g BCVFA/head cow/day [102]. In addition, β-sitosterol is essential for milk fat synthesis in BMECs in the concentration range of 0.1-10 µmol/L, and the potential pathways are JAK2/STAT5 and mTOR signaling pathways [103].

In vivo, there are also many studies on increasing milk production by adding relevant fatty acids to the feed (Table 3). Compared to the addition of stearic acid, the addition of 2% palmitic acid to the feed increased milk fat concentration (+0.11%), milk yield (+0.09 kg/d), and 3.5% fat-corrected milk yield (+1.9 kg /d) in Holstein cows [104]. In a study, adding palmitic acid (C16:0) and changing the dietary n-6/n-3 fatty acid (FA) ratio increased milk production in Holstein dairy cows [105]. In several studies, the addition of linseed to feed was shown to improve milk production and milk quality [106,107,108]. Supplementation of soybean oil or linseed oil 20 mL/d to lactating Anglo-Nubian goat diets, respectively, increased milk production (+15% and 11.7%) [109]. A study confirmed that adding plant oils (2% rapeseed oil, 2% peanut oil and 2% sunflower oil) to the basal feed positively affected milk production. The addition of rapeseed oil had the most significant effect [110]. Adding fish oil can increase the milk production of Holstein cows [111]. It was demonstrated that adding 2% dietary fish oil increased milk production (+2.23 kg/d) in Polish Holstein-Friesian cows [112]. In addition, microalgae rich in polyunsaturated fatty acids are also of interest. Studies have shown that adding microalgae to the diet of ruminants can improve the fatty acid content of milk and dairy products. Dietary addition of 20 g *Schizochytrium* spp./ewe/day improved the fatty acid profile of milk [113]. Dietary intake of 10 g/head/day of *Schizochytrium limacinum* in goats significantly increased lactogenic acid and docosahexaenoic acid fatty acid concentrations in milk [114]. The inclusion of docosahexaenoic acid-rich microalgae (1.22 to 2.90 g/kg dry matter) in the feed may be beneficial for milk, fat-corrected milk and energy-corrected milk yield [115].

Therefore, we found a lot of literature to support that fatty acids promote the synthesis of milk yield, which is of great significance for improving milk yield. At the same time, we can learn that some fatty acids can also promote the proliferation of related cells and the synthesis of milk fat. Most fatty acids, such as acetic acid, oleic acid, palmitic acid, etc., have a positive effect on the regulation of milk fat synthesis. They can stimulate the expression of genes related to milk fat syntheses such as SREBP1, PPARG and FABP3. In addition, mTOR and JAK2-STAT5 signaling pathways still play an active role in promoting milk fat synthesis by β-sitosterol.

**Table 3 animals-13-00419-t003:** Effects of fatty acids on milk production and its potential regulatory pathways.

Items	Treatments	Functions	Potential Signaling Pathways	References
Acetate (+)	Bovine mammary epithelial cells (6 mmol/L)	Milk fat synthesis ↑ Cell proliferation ↑	mTOR/eIF4E signaling pathway	[97]
Acetate, β-hydroxybutyrate and their interaction (+)	Dairy cow mammary epithelial cells	Triglyceride contents ↑ Lipid droplet formation ↑	SREBP1 signaling	[98]
Long Chain Fatty Acids (+)	Goat mammary epithelial cells	PPARG expression ↑	——	[99]
Branched-chain volatile fatty acids (+)	Chinese Holstein cows (60 g BCVFA per cow per day)	Milk fat synthesis ↑	——	[102]
β-sitosterol (+)	Bovine mammary epithelial cells (0.1 to 10 μmol/L)	β-casein synthesis ↑	JAK2/STAT5 and mTOR signaling pathways	[103]
Oleic acid, stearic acid, and palmitic acid (+)	Dairy cow mammary epithelial cells	Triglyceride contents ↑	FABP3-SREBP1/PPARG signaling	[100]
Oleic acid, linoleic acid, and linolenic acid	Bovine mammary epithelial cells (The ratio is 2:13.3:1)	Fat and protein synthesis ↑	——	[101]
Palmitic acid (+)	Holstein cows	Milk production ↑ Milk fat ↑	——	[102]
Palmitic acid, n-6/n-3 fatty acids (+)	Holstein cows	Milk production ↑	——	[103]
Linseed (+)	Cilentana dairy goats (20%)	Milk production ↑	——	[108]
Linseed (+)	Italian Friesian dairy cows (700 g/head/d)	Milk production ↑	——	[107]
Soybean, flaxseed oils (+)	Anglo-Nubian goats	Milk production ↑	——	[109]
Linseed, verbascoside, vitamin E (+)	Lacaune ewes	Milk production ↑	——	[106]
Rapeseed oil (+)	Holstein cows	Milk production ↑	——	[110]
Fish oil (+)	Polish holstein-friesian cows	Milk production ↑	——	[112]
Fish oil (+)	Holstein cows	Milk production ↑	——	[111]
Microalgae Schizochytrium spp. (+)	Crossbred dairy ewes [Lacaune x Local (Greek) breed]	Improves milks’ fatty acid profile	——	[113]
Schizochytrium limacinum marine algae (+)	Multiparous Alpine goats	DHA and rumenic acid concentration↑	——	[114]
Docosahexaenoic acid-rich microalgae (+)	Holstein cows	Improves milks’ fatty acid profile	——	[115]

↑: increase; ↓: decrease; ——: reported but not fully confirmed.

## 4. The Enhancement of Milk Production by Carbohydrates

Carbohydrates are tightly related to energy metabolism, and glucose is one of the most important carbohydrates. It is an essential precursor for lactose synthesis in the lactating mammary gland, providing fuel for milk synthesis and immune function in dairy cows [116]. The utilization of glucose by the mammary gland includes (1) synthesis of lactose; (2) generation of nicotinamide adenine dinucleotide phosphate (NADPH); (3) synthesis of milk fat; (4) production of energy; (5) synthesis of nucleic acids and amino acids [117]. In the mammalian mammary gland, two independent transport mechanisms are involved in glucose uptake: a facilitative transport mediated by the glucose transporter family and sodium-dependent transport mediated by the sodium+/glucose co-transporter [118]. The most important in glucose-promoted milk protein synthesis is the AMPK-mTOR pathway (Table 4). There is an interaction between mTOR and AMPK signaling pathways [119]. The adenosine monophosphate (AMP)-activated protein kinase (AMPK) can maintain energy balance under metabolic stress at the cellular and physiological levels and is closely related to intracellular energy metabolism [120]. AMPK is an upstream factor of mTOR. When AMPK is activated by other factors, mTOR signaling pathway is inhibited, leading to a reduction in the synthesis of related proteins [121]. 

The previous study has shown that the concentration of total amino acids in milk is highest when 60 g/d glucose is administered to lactating goats. During this process, added glucose may regulate the utilization of amino acids and the synthesis of related proteins through the AMPK-mTOR pathway [122]. In BMECs, adding glucose can regulate casein synthesis through the AMPK/mTOR signaling pathway [55]. In addition, adding chitosan to feed can alter rumen fermentation and increase milk production in lactating dairy cows [123].

It can be seen that there are not many studies on the effect of carbohydrates on lactation, but the studies show that glucose and chitosan can promote the synthesis of related proteins and milk production in dairy cows. They are linked to energy-related genes, including AMPK. As an upstream factor of mTOR, AMPK acts through the AMPK/mTOR pathway under the stimulation of glucose.

## 5. The Enhancement of Milk Production by Other Chemicals

In addition to the main nutritional factors related to milk protein synthesis, other chemicals—including phenolic compounds and lactation-related hormones (prolactin, estrogen, growth factors) [124]—also affect the synthesis of milk components (Table 5). Phenolic compounds have aromatic rings, and at least one hydroxyl group and phenolic derivatives can be divided into flavonoids and non-flavonoids. A study has shown that the 20 μmol/L daidzein (an isoflavone extract from soy)-treated BMECs can promote casein production and stimulate milk fat synthesis-related protein synthesis and cell proliferation through ERa-dependent NF-κB 1 activation [125]. Adding phenolic compounds extracted from *pistacia lentiscus* to BMECs can also increase milk content [126].

Several hormones, alone or in synergy, can play a crucial role in milk protein synthesis. Hormones generally act on breast cells to promote cell growth, promoting the synthesis of related milk proteins [124]. In the process of hormone-promoted milk synthesis, the related pathways still have mTOR and JAK-STAT signaling pathways (Table 5). The JAK-STAT signaling pathway is related to the transcriptional level and regulates the synthesis of mammary milk proteins, while the mTOR pathway affects the synthesis of related proteins at the translational level.

Prolactin is a polypeptide hormone. Prolactin can interact with specific receptors on mammary epithelial cells. Its receptors are located on the cell membrane, and the receptor sites on the membrane have a high affinity for hormones [127]. Estrogen is a steroid hormone, and its receptors include estrogen nuclear receptors and estrogen membrane receptors. It is also closely related to prolactin content and progesterone in breast cells [128]. Prolactin can stimulate the expression of LAT1 to increase milk protein synthesis [129]. The LAT1, also known as SLC7A5, is part of the SLC family and is closely related to amino acid transport [130]. Prolactin can activate Tudor staphylococcal nuclease (Tudor - SN), thereby regulating milk protein synthesis [131]. Estrogen-stimulated milk fat synthesis has been demonstrated in BMECs. The interaction between methionine and estrogen can promote milk fat synthesis and positively regulate SREBP-1c expression through FABP5 [29]. As previously described, FABP5 and SREBP-1c are associated with fatty acids promoting milk fat synthesis. The interaction between prolactin, estrogen and amino acids can encourage cell and milk synthesis. The validated pathways are AnxA2-mediated PI3K-mTOR-SREBP-1c/Cyclin D1, GRP78/mTOR, and U2AF65-mediated mTOR-SREBP-1c signaling pathway [30,70,71]. Annexin A2 (AnxA2) can interact with hormone receptors to increase the phosphorylation of PI3K, thereby affecting its downstream pathways [132]. Prolactin and epidermal growth factor (EGF) stimulate β-casein synthesis in mouse mammary epithelial cell line HC11 through the PI3-kinase/Akt/mTOR signaling pathway [133].

In addition, studies have shown that sodium butyrate can activate G protein-coupled receptor 41 (GPR41) and its downstream signaling pathways [134]. Sodium butyrate often replaces unstable butyric acid. GPR4 is associated with lipid metabolism and can be activated by short-chain fatty acids [135,136], thereby promoting the AMPK/mTOR/S6K signaling pathway and increasing the nuclear translocation of SREBP1 to promote milk fat synthesis in BMECs. Camellia seed oil and all-trans retinoic acid can promote casein and milk fat synthesis in BMECs, and the mTOR and JAK2/STAT5 signaling pathways can explain the mechanism [137,138]. Supplementation of 120 mg/500 kg of folic acid in dairy cows can increase milk production in peripartum cows [139]. Adding 20 mg GSPE (grape seed proanthocyanidin extract) /kg body weight/d to the diet can significantly improve the milk production of early lactating dairy cows [140]. Lutein-feeding lactating Holstein cows can promote lactose synthesis and metabolism, which may provide clues for the mechanism of lutein regulation of lactation in dairy cows [141]. In brief, phenolic compounds, lactation-related hormones and other chemicals can enhance milk synthesis. The mTOR pathway remains essential in these studies. These compounds can either activate the synthesis of related enzymes to enhance milk protein synthesis or stimulate factors related to the mTOR pathway.

**Table 5 animals-13-00419-t005:** Effects of other chemicals on milk production and its potential signaling pathways.

Items	Treatments	Functions	Potential Signaling Pathways	References
Daidzein (+)	Primary bovine mammary epithelial cells (20 µmol/L)	α- and β-casein ↑ Lipid synthesis ↑ Cell amount ↑	ERα-dependent NFƘB1 signaling	[125]
Polyphenols from lentisk ethanolic extract (+)	Bovine mammary epithelial cells	Lactose synthesis ↑ Secretion of whey proteins ↑ Casein contents ↑	——	[126]
Prolactin (+)	Bovine mammary epithelial cells	Milk protein synthesis ↑	LAT1 signaling	[129]
Prolactin (+)	Bovine mammary epithelial cells	Milk protein synthesis ↑ Tudor-SN expression ↑	——	[131]
Estrogen (+)	Bovine mammary epithelial cells	Milk fat synthesis ↑	FABP5/SREBP-1c signaling	[29]
Estrogen or prolactin (+)	Bovine mammary epithelial cells	Milk synthesis ↑ Cell proliferation ↑	PI3K-mTOR-SREBP-1c/Cyclin D1 signaling pathway	[70]
Estrogen and prolactin (+)	Bovine mammary epithelial cells	Milk protein synthesis ↑ Milk fat synthesis ↑ Cell proliferation ↑	GRP78/mTOR signaling pathway	[30]
Prolactin and β-estradiol (+)	Bovine mammary epithelial cells	β-casein synthesis ↑ Triglyceride synthesis ↑, Lactose synthesis ↑; Cell proliferation ↑	U2AF65/mTOR-SREBP-1c signaling pathway	[71]
Prolactin and epidermal growth factor (+)	Mouse mammary epithelial cell line HC11	β-casein ↑	PI3K/Akt/mTOR signaling pathways	[133]
Sodium butyrate (+)	Bovine mammary epithelial cells	Milk fat synthesis ↑	GPR41/AMPK/mTOR/S6K- SREBP1signaling pathway	[134]
Camellia seed oil (+)	Differentiated bovine mammary epithelial cells	β-casein ↑	PI3K-mTOR-S6K1 and JAK2-STAT5 signaling pathways	[137]
All-trans retinoic acid (+)	Bovine mammary epithelial cells	Casein synthesis ↑ Fatty acid composition ↑	JAK2/STAT5 pathway and downstream mTOR signaling pathway	[138]
Folic acid (+)	Lactating cows (120 mg/500 kg per cow)	Milk production ↑	——	[139]
Grape seed proanthocyanidin extract (+)	Holstein dairy cattle (20 mg GSPE/kg of body weight/day)	Milk yield ↑	——	[140]
Lutein (+)	Lactating Holstein cows	Milk lactose synthesis ↑	——	[141]

↑: increase; ↓: decrease; ——: reported but not fully confirmed.

## 6. STRING Database Analysis

In order to construct the interaction network between proteins and explore the core regulatory genes for milk production, we employ the STRING database (https://string-db.org/, accessed on 1 July 2022), a widely used database and online resource devoted to investigating functional protein association networks. We submitted 36 genes to the STRING database and specified the organism as cattle (*Bos taurus*). The result is shown in Figure 2.

The genes related to milk protein synthesis are widely investigated in milk production studies. The mammalian target of rapamycin (mTOR) is the most important regulator of these pathways. The mTOR is a class of serine/threonine kinases whose catalytic domains contain different phosphorylation sites [142]. The mTOR, together with its upstream genes (PIK3CB, MAPK1 and MAPK3) as well as its downstream genes (EIF4EBP1 and RPS6KB1), may enhance milk protein and nucleotide synthesis by inhibiting autophagy and other protein degradation pathway. Another complex pathway, the JAK2-STAT5A, may increase the RNA abundance of casein and thus enhance milk production. 

The genes related to fat synthesis and metabolism are also crucial in milk production studies. Milk fat is the primary energy component of milk, and fatty acids are the essential components for milk fat synthesis. Milk fat synthesis and regulation are currently known to rely on SREBP1 and PPARG. The genes networks analysis showed that fatty acid metabolism genes, including FABP3 and FABP5 are in crosstalk with other essential genes and pathways mainly through PPARG. FABP3 and FABP5 belong to a fatty acid-binding proteins (FABPs) family, and the members of FABPs have formed proteins ranging from 14 to 15 kDa, which could participate in the uptake and transport of long-chain fatty acids. FABP3 and FABP5 genes are enriched in the KEGG pathways related to metabolism and glucose/energy metabolism, and Gene Ontology annotations related to transporter activity and cytoskeletal protein binding.

The transport systems that transport nutrients across membranes also play essential roles in promoting milk production. The genes—including SLC1A5, SLC6A14, SLC38A2, and SLC7A5—are all proven to be enrolled in the milk production process. SLC1A5 can transport all neutral amino acids, including glutamine, asparagine, and branched-chain and aromatic amino acids. SLC6A14 transports all amino acids with exception of the acidic ones: aspartate and glutamate. SLC38A2 transports neutral amino acids, and SLC7A5 uptakes sizeable neutral amino acids such as phenylalanine, tyrosine, L-DOPA, leucine, histidine, methionine and tryptophan. These results indicate that amino acids are necessary for milk production. As the need for protein and other nutrient increase during lactation, we assume the expression levels of transporters will be upregulated. If researchers further focus on the expression of transporters during lactation, we may know more about the nutrient needed during lactation. The study of transporter expression during lactation will become even more critical as the nutrients needed to improve milk quality during lactation become better understood.

We note that research on nutrient supplements to promote milk production is gaining traction. However, the entire pathway for milk production is unclear. Applying bioinformatics methods to the lactation process may help build the entire pathway network for milk production; mainly, the bioinformatics methods help to understand the effect of different nutrients, including amino lipid carbohydrates and other nutrients, on lactation. A better understanding of the associated trophic factors and signaling pathways may help us optimize the trophic factor profile in animal feeding.

## 7. Conclusion and Future Perspective

This review summarizes the enhancement of milk production by nutrient supplements and the related pathways. Most works focus on enhancing milk production with amino acids and fats. In contrast, a relatively small amount of work focuses on the enhancement of milk production with other chemicals, such as carbohydrates, phenols, and hormones. We found that the nutrients may influence the pathways related to protein synthesis as well as fat synthesis and metabolism. Additionally, the transport systems that transport nutrients across membranes also play essential roles in promoting milk production. Of all the pathways, mTOR, JAK2-STAT5A, SREBP1, and PPARG were frequently involved. We speculate that these pathways play essential roles in promoting milk production. However, the entire pathway for milk production still needs to be clarified. Further research should be carried out to interpret the enhancement of milk production by nutrient supplements.

Moreover, most current works employ mammary epithelial cells to investigate the enhancement of milk production. It is supposed that the results from dairy animals are more convincing than that from the cell model. As the genetic base of dairy animals has changed compared with decades ago, we suggest that future studies can be conducted again with the current breeds of dairy animals. We encourage validating cell-level experiments at the animal level in subsequent studies.

## Figures and Tables

**Figure 1 animals-13-00419-f001:**
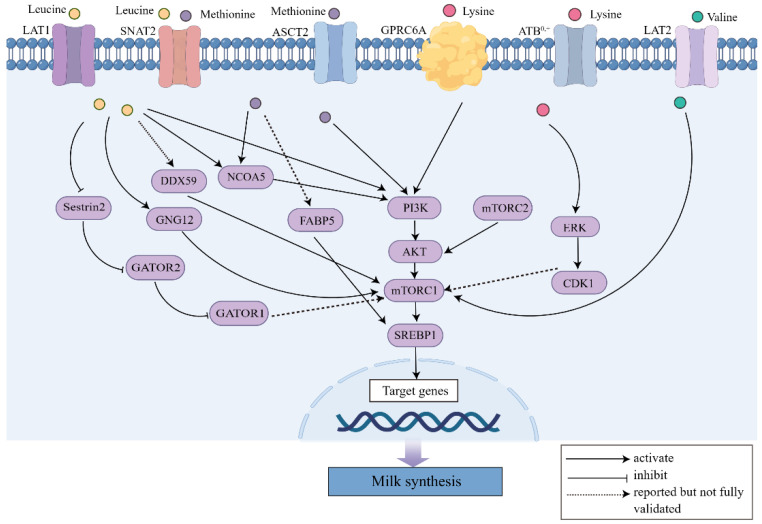
The regulation of milk synthesis by methionine, leucine, valine and lysine. The amino acids were transport across membrance by different transportors including LAT1, LAT2, SNAT2, ASCT2, and ATB^0,+^, respectively. GPRC6A: G protein-coupled receptor class C group 6 member A, is a basic amino acid receptor. The mammalian target of rapamycin complex 1 (mTORC1) is the most important regulator of these pathways. The activation of mTORC1 increases milk protein synthesis and regulates milk fat synthesis.

**Figure 2 animals-13-00419-f002:**
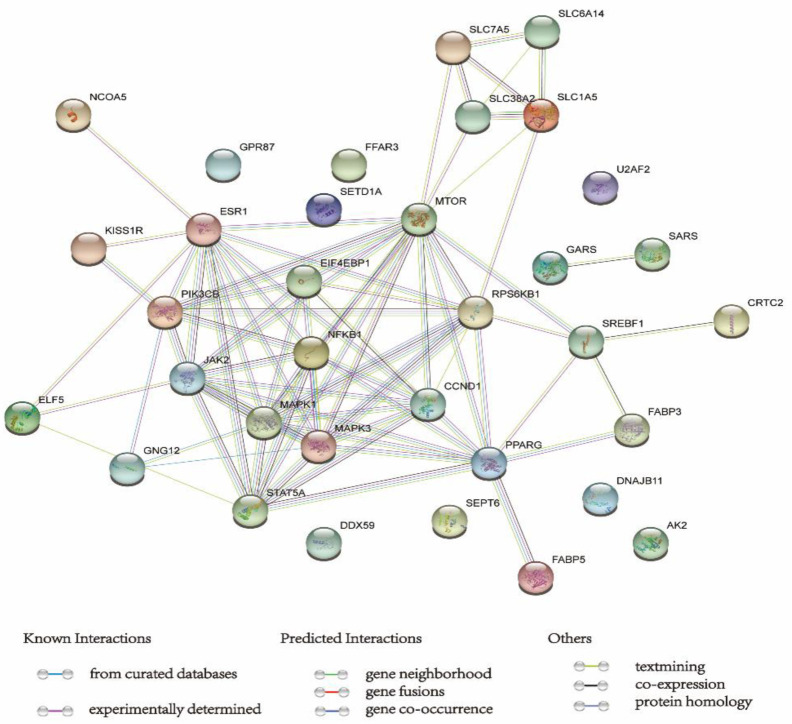
Network of lactation-related proteins under nutritional supplementation.

**Table 2 animals-13-00419-t002:** Effects of peptides on milk production and its potential regulatory pathways.

Items	Treatments	Functions	Potential Signaling Pathways	References
Ghrelin (+)	Mature Guanzhong Saanen dairy goats and mammary epithelial cells	β-Casein synthesis ↑	——	[77]
Ghrelin (+)	Primary bovine mammary epithelial cells	β-Casein synthesis ↑	ERK1/2 and AKT signaling pathways	[78]
Kisspeptin-10 (+)	Bovine mammary epithelial cells (100 nmol/L)	β-Casein Synthesis ↑	CSN2 via GPR54 and its downstream signaling pathways mTOR, ERK1/2, STAT5 and AKT	[81]
Methionyl-methionine dipeptide (+)	Bovine mammary epithelial cells (80 µg/mL)	PepT2 expression ↑ β- Casein synthesis ↑	AK2-STAT5 and mTOR signaling pathways	[84]
Threonyl-phenylalanyl-phenylalanine (+)	Bovine mammary epithelial cells (5, 10 and 15%)	PepT2 mRNA abundance↑	——	[85]
Threonyl-phenylalanyl-phenylalanine (+)	Bovine mammary epithelial cells (add lactogenic hormone treatment)	PepT mRNA abundance↑	——	[86]
Octopus peptide (+)	Mouse mammary epithelial cell line	β- Casein synthesis ↑ Cell proliferation ↑	——	[87]

↑: increase; ↓: decrease; ——: reported but not fully confirmed.

**Table 4 animals-13-00419-t004:** Effects of carbohydrates on milk production and its potential regulatory pathways.

Items	Treatments	Functions	Potential Signaling Pathways	References
Glucose (+)	lactating dairy goats (60 g/d)	Amino acid ↑	AMPK-mTOR signaling pathways	[122]
Glucose (+)	Bovine mammary epithelial cells	β-casein ↑ Cell proliferation ↑	AMPK/mTOR signaling pathways	[55]
Chitosan (+)	Holstein cows (225 mg/kg bodyweight)	Milk yield ↑ Fat-corrected milk, protein and lactose production ↑	——	[123]

↑: increase; ↓: decrease; ——: reported but not fully confirmed.

## Data Availability

Not applicable.

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
