# Peer review of "Enhancing Milk Production by Nutrient Supplements: Strategies and Regulatory Pathways"

_animals, 2023, doi:10.3390/ani13030419_

Round 1

Reviewer 1 Report (New Reviewer)

This review summarize how different types of organic compounds influence on metabolic pathways in relation to milk production, based on literatures and lactation-related protein network analysis performed by authors. It is written comprehensively and likely to arouse interests of those who are working in dairy area, but there are some confusions and questions as indicated below. English proof reading should be done by experts familiar with the research area.

  • In section 2, there is confusion on milk protein and lipid synthesis pathways. Firstly, the roles of mTORC1 and mTORC2 in association with other pathways, such as PI3K/Akt, should be well explained in terms of milk protein and lipid syntheses. For this purpose, it is strongly recommended to illustrate both pathways as figure(s).
  • Abbreviations should be spelled out at the first place, e.g., GPCR appears first in subsection 2.4. but it is spelled out in subsection 2.5.
  • Causal relationship between enhancement of milk yield and stimulation of biosynthetic pathways of milk-related compounds is unclear. Authors explain that enhancement of BMEC proliferation by such compounds evokes increasing milk production, but is that all? How do you think about influence of such compounds on cell development or on osmotic pressure in relation to milk yield?
  • Typos should be checked throughout the manuscript (e.g., PPARG).
  • In section 6 and other, “chemists” should be “chemicals”.
  • In section 6, Does “hydroxyl bond”  mean “hydroxyl group”?
  • In section 6, what is daidzein? Explanation should be added.
  • In section 6, Pistacia lentiscus should be in Italic.
  • In section 7, how much is the STRING database analysis reliable? How can we know the accuracy of the analysis results?
  • In section 7, “The mTOR, …degradation pathway.” How does it happen?

Author Response

Reviewer 2 Report (New Reviewer)

The study is presenting a very interesting summary of studies on influence of the nutritional supplements to enhance the milk production and milk quality in dairy cows. I recommend the work for publication after the following points are addressed by the authors:

1.     Simple summary is not well written. The justification of the study is not clear. The last two sentences should probably be merged. “The world's milk production reached 928 million tons in 2021. There is a huge market for milk and dairy products, and thus significant efforts are needed to enhance milk production. We summarize the enhancement of milk production. By nutritional supplements and related essential pathways on dairy animals.

2.       Page 3 line 147. What is “Ras GTPase and Raf, MEK and ERK

3.       Page 4 line 148. What is “L-valine

4.       Lie 152. What is “BCAAs

5.       Line 178 What is “ALA

6.       Line 187: please rewrite the sentence ”In recent years, researchers have focused their attention on the cellular level in order to explore the mechanisms of how this works

7.       Line 188: Please remove “on” in “Amino acids can positively affect on the proliferation and milk protein synthesis of mammary epithelial cells.

8.       Line 191: please do not start the sentence with an “and”

9.       Line 194 and 195: the sentence is not clear.

10.   Line 216: what does it mean? “They play important roles through the ERK1/2 and AKT signaling pathways

11.   Line 223: space is missing “(Methionyl-Methionine)positively

12.   Does it make sense to refer to Tables more often?

13.   Line 256: please rewrite the sentence” We can still find that researchers are exploring the effects of fatty acids on pathways related to milk fat synthesis at the cellular level.” Why still?

14.   Line 272: what is “0.1-10 μ M

15.   Line 288. Sentence can be improved “Therefore, we found a lot of literature to support that fatty acids promote the synthesis of milk yield, which is of great significance for improving milk yield.”

16.   Line 310: is it a mistake “5’-monophosphate”?

17.   Line 325: the sentence “Besides, chitosan can also promote milk production in dairy cows.” seems like a repetition

18.   Line 425: add “to” in “help understand

19.   The section conclusion needs language proofreading. Many sentences are unclear and poorly written

Author Response

Reviewer 3 Report (New Reviewer)

Overall, the manuscript is clear and well-written and provides concise and precise updates on the latest progress made in each area of research. The author highlights the enhancing roles of certain nutritional supplements including related amino acids, peptides and lipids and relevant pathways for the enhancement of milk production. This manuscript is well-addressed important structure of a “review” such as summarization/classification, and analysis. The table and visualization are understandable and meaningful. A summarization of relevant findings from previous articles of amino acids and peptides are also presented in Table 1 & 2.  An analysis information of certain targeted genes and milk and fat production (Line 383 and onwards, STRING database analysis) was presented and could be helpful for understanding (Figure 1).

Author Response

Reviewer 4 Report (New Reviewer)

General remarks

The manuscript’s topic is interesting for the scientific community, but I did find any problems.

Additional remarks:

In the manuscript, the regulatory pathways were more dominant compared to different strategies of the nutrient supplements.

Simple abstract was very short, please improve this section!

I suggest combining No 2 and 3 (amino acids and peptides) sections, and use subsections (2a – amino acid; 2b – peptides)!

No 2 section: feeding different supplements: no data whether authors protected or unprotected amino acid treatments were used?

Line 152: what does BCAAs mean? No 50 report: deal with sows not cows! See the reference list!

2.4 subsection: did not find any reports belonging to lysine supplementation?

Line 178: what does ALA mean?

Lines 184-185: interesting phrase, but folic acid is not amino acid!

3. section: did not find any reports belonging to the peptides supplements?

Lines 274-287: very dominates the linseed supplementation. Why not evaluate the microalgae, such as sweet water and marine algae? Suggested reports:

https://doi.org/10.3390/ani11041097

https://doi.org/10.3390/foods11192950

Line 315: “concentration of total amino acids is highest…” but where? In milk? Please clarify it!

To sum up, in this manuscript the regulation pathways are more dominant, moreover, feeding strategies are missing.

Round 2

Reviewer 1 Report (New Reviewer)

Legend of Figure 1 should be improved. For example, there is no explanation for the channel protein. Furthermore, is only one channel protein importing all the amino acids? Meaning of background colors of proteins also should be explained. "Please" in the legend should be removed. What is the meaning of different thickness of arrows? What is the meaning of the arrows? Activation?  Is there any suppression regulation in those pathways?  

Author Response

Reviewer 2 Report (New Reviewer)

Dear authors,

 The manuscript needs improvement of the text.

 1.       Simple summary and conclusions sections are still not well written. Do not please refer to researchers so much. Better to refer to studies, to research but not researchers. Especially  strange this sounds: “researchers treated mammary epithelial cells or fed the animals with appropriate nutrients”. Please rewrite the sections.

2.       My comment 12. “Does it make sense to refer to Tables more often?” is misunderstood. I suggested to refer to the tables more often since you make conclusions based on the information in the table.

Author Response

Reviewer 4 Report (New Reviewer)

All sections of this manuscript were improved according to instructions, so I recommend this manuscript for publishing in the "Animals" journal.

Author Response

Thank you for your review, we have revised based on comments from other reviewers

This manuscript is a resubmission of an earlier submission. The following is a list of the peer review reports and author responses from that submission.

Round 1

Reviewer 1 Report

The paper “A review on the promotion milk production by nutrient supplement: the strategy and the regulatory pathway” summarize how some nutritional factors can influence milk production and his components. They also tried to characterize and identify the most important regulatory pathways in this contest with the objective to highlight the factors influencing milk synthesis.

I think that authors elaborated on the literature; indeed, their report is very accurate, even if it is written like an introduction of a degree thesis.

However, I have some concerns about the paper.

1)      It’s not clear to me how this review can help to improve milk production. They wrote in L40-42: “However, the average milk yield of cow’s ranges from 1.3 tons to 10 tons in different countries. The average milk production of dairy cows in Saudi Arabia and Israel is more than 10 tons, while India is only 1.3 tons [4]. Therefore, it is extremely important to increase the milk production of each cow”. Clarify better the objective and the prospective of this work.

2)      There is a lack of references (and to the topic) about genetic breeding and livestock management to improve milk production especially in developing country.

3)      Lines 87-96: the paragraph is out of topic. I think could be removed.

4)      In many cases it’s not explicit if the cited experiments to prove the role of nutrients factors are in vitro or in vivo or in which species are conducted (human, bovine, pig).

5)      L78-86: ???

6)      L 97-98: “The classical pathway related to amino acid regulation of milk protein synthesis is PI3K/AKT/mTOR (Table 1)”. I couldn’t find in the table.

7)      L 255- 264: the paper is a review not a chapter of a scholar book. It’s wrote for experts not for students.

8)      Rewrite conclusion and future prospective.

9)      In particular (L 398-401), why did authors present STRING analysis in a conclusion of the review? How did they perform the analysis? I think it could be better to write a small “Mat and Met” paragraph to clarify the procedure. Which was the protein list that authors used, and which organism they considered (human, bos taurus……)?......

10)   L 411-442: the conclusion is derived from the literature reviewed or from STRING analysis. It’s not clear for me.

11)   L 435-442: to rewrite

Reviewer 2 Report

There are many fundamental errors in this review. The title and the content do not match and authors haven't focused in nutrient supplementation and general description has given in relation to nutrients and how those nutrients would  influence on milk composition and yield. Authors haven't focused on dairy animals and sometime discuss about human milk as well. 
